# Advances in *Pasteurella multocida* Vaccine Development: From Conventional to Next-Generation Strategies

**DOI:** 10.3390/vaccines13101034

**Published:** 2025-10-07

**Authors:** Adehanom Baraki Tesfaye, Geberemeskel Mamu Werid, Zhengyu Tao, Liuchao You, Rui Han, Jiayao Zhu, Lei Fu, Yuefeng Chu

**Affiliations:** 1State Key Laboratory for Animal Disease Control and Prevention, College of Veterinary Medicine, Lanzhou University, Lanzhou Veterinary Research Institute, Chinese Academy of Agricultural Sciences, Lanzhou 730000, China; adehanommek@gmail.com (A.B.T.); tzylzu@126.com (Z.T.); 82101221298@caas.cn (L.Y.); 13040541018@163.com (R.H.); 18993462001@163.com (J.Z.); 2Gansu Province Research Center for Basic Disciplines of Pathogen Biology, Lanzhou 730046, China; 3Tigray Agricultural Research Institute, Mekelle, P.O. Box 492, Ethiopia; 4Australian Centre for Antimicrobial Resistance Ecology, School of Animal and Veterinary Sciences, Roseworthy, SA 5371, Australia; gebremeskelmamu.werid@adelaide.edu.au; 5Davies Livestock Research Centre, School of Animal and Veterinary Sciences, Roseworthy, SA 5371, Australia

**Keywords:** Antigen composition, *Pasteurella multocida*, vaccines, virulence gene

## Abstract

*Pasteurella multocida* is a Gram-negative bacterium causing significant livestock diseases, like fowl cholera and hemorrhagic septicemia in cattle, and wound infection in humans. Classified into four subspecies and five capsular serotypes, it possesses multiple virulence factors, including capsular polysaccharides (CPSs), lipopolysaccharides (LPSs), outer membrane proteins (OMPs), iron acquisition proteins, and toxins that serve as vaccine targets. Antimicrobial treatment is challenging, so vaccination is key. Commercial vaccines include killed and live attenuated types, which are commonly used, though they have intrinsic problems. Advanced vaccines like recombinant subunit and DNA vaccines are emerging. Subunit vaccines targeting OMPs (*OmpH, OmpA, PlpE, VacJ,* and *PmSLP)* and recombinant *Pasteurella multocida* toxin (rPMT) show high efficacy in animal models, and their recombinant proteins induce strong immune responses. DNA vaccines have promise but limited use. The challenges in vaccine development are the strain diversity, short-term immunity, and inconsistent cross-protection. There is also a lack of research on recombinant and subunit vaccine development for small ruminants. Future research should focus on multivalent vaccines, optimization, including improving adjuvants and optimizing DNA vaccine delivery.

## 1. Introduction

Pasteurella species are facultatively anaerobic, Gram-negative bacteria, coccobacillary- to rod-shaped organisms that belong to the family *Pasteurellaceae* [1]. There are multiple species of Pasteurella, with *Pasteurella multocida* being the most clinically significant since it infects a wide host range, including domestic and wild mammals, birds, and reptiles [2]. Pasteurella is a disease of economic importance to global animal husbandry that causes fowl cholera, hemorrhagic septicemia (HS), and respiratory disease in ruminants, progressive atrophic rhinitis (PAR) and pneumonic pasteurellosis in swine, snuffles in rabbits, lower respiratory infection in ungulates, and opportunistic infections due to bites from pets or scratch wounds in humans [3,4,5].

*P. multocida* can be classified into four subspecies, namely, *P. multocida multocida*, *P. multocida gallicida*, *P. multocida septica* [1,6], and *P. multocida tigris* [7]. Additionally, *P. mutolocida* was further categorized into five capsular serotypes (A, B, D, E, and F) and sixteen somatic serotypes (1–16) [8]. Among these serotypes, B2 and E2 are known to cause hemorrhagic septicemia, and they may lead to potential cases of pneumonia, enteritis, or septicemia caused by various capsular serogroups and somatic serotypes [9].

Various somatic serotypes (A, B, D, and E) cause hemorrhagic septicemia in buffalo, cattle, pigs, elephants, bison, goats, and mithun, whereas D, B, A, and A/D in pigs, chickens, buffalo, and turkeys cause respiratory diseases, such as bronchopneumonia, pneumonia, and atrophic rhinitis. Serotypes A and B in poultry and bovine cause fowl cholera and hemorrhagic septicemia, respectively. Serotype F affects sheep and cattle [10,11]. *P. multocida* type A plays a significant role in the respiratory infection of sheep [12].

Most capsular serogroups are associated with specific diseases. For instance, strains of hemorrhagic septicemia are related to capsular serotypes B and E, fowl cholera strains are typically classified as capsular serotype A, and atrophic rhinitis strains usually fall under capsular serotype D [13]. Types B and E are the main causes of bovine hemorrhagic septicemia, which causes substantial annual damage to livestock globally, especially in low- and middle-income countries [14].

These serotypes are the main determinant of hosts and diseases: for example, types B2 and E2 cause bovine hemorrhagic septicemia, serogroup A (specifically A, A3, and A4) is the main cause of fowl cholera, and subgroups D and A are linked to F porcine atrophic rhinitis. Among these, *P. multocida* is a leading human pathogen, especially in infected wounds from bites of pet animals from serogroup D [15].

Typically, manifested disease outbreaks are associated with a respiratory disease complex, and their impact is observed at the individual-animal level. Globally, *P. multocida* has been isolated from multiple host species of origin [9,10]. *P. multocida* is a strict opportunistic bacterium that resides in the nasopharyngeal and oral mucous membranes. Pasteurellosis can be caused by *P. multocida,* occurring as more or less acute septicemia and causing signs associated with the primary infection organ. Local and systemic defense mechanisms can be impaired by factors such as prolonged transportation, overcrowding, climate change, or respiratory viral infections [16,17].

Due to the widespread use of antimicrobials for prophylaxis and curative treatment, the emergence of antimicrobial resistance genes in Pasteurellosis in food-producing animals requires the development of rapid diagnostic tests and effective vaccines to mitigate the microbial burden and reduce antimicrobial dependence [18,19,20,21,22,23]. Even though biosecurity measures have contributed to reducing the spread of *P. multocida*, immunization methods are the most potent preventive measure [5,24].

Despite the availability of multiple *P. multocida* vaccines, their efficacy remains inadequately studied due to a lack of well-structured studies and significant variability in reported outcomes, making it challenging to derive a reliable pooled protection rate estimate [25]. According to Mostaan et al. [26], multiple formulations of vaccines were developed to protect against *P. multocida* bacteria, but no single vaccine could fully control all the serotypes. Strategies for vaccine design that result in improved cross-protection vaccines represent the most effective means of control. There has been a growing effort to develop vaccines that are more potent and cause less damage. Existing vaccines, including inactivated and live vaccines, have intrinsic problems related to their safety, efficacy, and immunity duration. So far, the new generation of vaccines have also been considered to bring good progress in disease control, and may soon be an option to cope with the drawbacks of killed and live vaccines. Therefore, a review was undertaken to explore the available *P. multocida* vaccine types, development strategies, and components in various hosts, along with their safety and efficacy.

## 2. Immunogenic Components of *Pasteurella multocida*

The pathogenesis of *P. multocida* is shaped by numerous virulence factors. These include genes associated with capsule formation, lipopolysaccharides (LPSs), fimbriae and adhesins, toxins, proteins involved in iron-controlled mechanisms and iron acquisition, components related to sialic acid metabolism, hyaluronidase, and outer membrane proteins (OMPs) [27,28]. Several hemin- and hemoglobin-binding proteins also contribute to the virulence. None of them was protected in mouse trials under *P. multocida* experimental infection when inoculated as a single formulation. These hemin- and hemoglobin-binding proteins require other combination strategies to produce vaccines against pathogens that present several alternative iron-uptake proteins [29]. The functions of virulence genes and major virulence genes are shown in Table 1.

Components within the bacterial outer membrane, especially transmembrane proteins and lipoproteins, have a crucial role in the pathogen’s interaction with the host environment and in pathogenesis and immunity. OMPs include several functional groups, such as transporter proteins, structural proteins, binding proteins, adhesins, putative adhesins, membrane-associated enzymes, and protein assembly machinery. OMPs comprising porins are potentially immunogenic, where they are conserved within bacterial phylogeny, rendering them attractive as vaccine candidates [30].

In general, *P. multocida* type A expresses capsules consisting of hyaluronic acid that are involved in adherence to immune cells [31], whereas type B contains a polysaccharide capsule consisting of galactose, arabinose, and mannose sugar residue, as well as a type F chondroitin or type D heparin/heparin surface [8,32].

**Table 1 vaccines-13-01034-t001:** Pasteurella multocida virulence factors and associated genes.

Virulence Factors	Corresponding Virulence Genes	References
Adehesins and fimbriae	*PtfA*, *pfhA*, *fimA*, *tadD*, *hsf-1*, *hsf2*	[2,16,20,22,28,33,34,35]
Toxins/dermonecrotic	*toxA*	
Iron acquisition	*ExbB*, *exbD*, *tonB*, *hgbA*, *tbpA*, *hgbB*, *fur*	
Sialidases/neuraminidase	*nanH*, *nanB*	
Hyaluronidase	*pmHAS*	
Protections/outer membrane proteins	*ompH*, *oma87*, *ompA*, *plpB*, *plpB*, *psl*	
Superoxide dismutase	*sodA*, *sodC*	

### Capsular Serotype and Virulence Genes

*P. multocida* infects a wide range of hosts. Expressions of different capsular serotypes and virulence genes may show differences across different animals. Accordingly, capsular type A strains had a high level of adaptation to bovine and poultry species. In swine, there were *capA* and *capD* strains. A high number of *capD*-positive strains were found in small ruminants. Additionally, *capF* was found on wild-type strains isolated from diseased cattle. From these reports, none of the isolates incorporated *capE*, while all strains of *capB* originated solely from buffaloes. Even though there is a significant difference in proportion, most combinations of genes encoding outer membrane proteins, colonization factors, iron acquisition factors, and superoxide dismutase have no basic differences regarding host specificity. In general, virulence genes do not exhibit host specificity [2,11,23,35,36,37]. The frequency of isolation capsular serotype versus host relationship is shown in Figure 1.

Capsular types A, B, and D commonly occur in domestic ruminants, and a high occurrence of the *toxA* gene and capsular D isolate was noticed in small ruminants [36]. Studies investigated the proportions of capsular types and their virulence-associated genes; specifically, there were 22 important genes in *P. multocida* (*pmHAS*, *tbpA*, *hgbA*, *hgbB*, *sodA*, *sodC*, *oma87*, *ompH*, *plpB*, *ptfA*, *fimA*, *hsf1*, *hsf2*, *pfhA*, *tadD*, *nanB*, *exBD-tonB*, *nanH*, *exbB*, *fur*, *toxA*, and *ompA)*, almost all of which were reliably present in domestic ruminants [8,34,36].

Significant relationships were observed between virulence genes and their relevant disease epidemiology, such as gene markers, e.g., the *toxA*, *tbpA, hgbB,* and *pfhA* genes. Swine disease was found to be in association with the *toxA* gene, whereas *pfhA* was associated with bovine diseases [2]. *tbpA* and *nanH* were found in 100% of nasal swabs and lung tissue in cattle [51]. *tbpA* and *toxA* genes played a considerable role in the disease epidemiology of sheep respiratory disorders [12,52]. The isolation proportions of *toxA* and *tbpA* genes from goats showed similarity to sheep isolates. This comparison of virulence gene profiles shows the probability of *P. multocida* transmission between sheep and goats [39,53]. Regarding the bovine respiratory disease complex, prevalent isolates were those containing *pfhA*, *tbpA,* and *capA*. On the other hand, in sheep, the dominant isolates were those with *tbpA*, *toxA*, and *capD* [23]. In sheep and goat, *P. multocida* capsular type A was the most frequently isolated, followed by type D. Additionally, the high isolation frequency of *tbpA*, followed by the *pfhA*, *toxA*, and *hgbB* genes, showed probable significance in the pathogenesis in sheep and goats [43,48,49,54]. The *toxA* gene was detected in sheep with pneumonia and was found to be toxigenic [55].

Several reports involving the capsular serotypes versus host relationships were described. Two capsular serogroups, types A and D, are common capsular types, and the *toxA* gene is an important marker gene for describing the pathogenic potential of *P. multocida* strains in swine [34]. In poultry, type A was predominant [35,36,37,44,56,57]. Similarly, in cattle, even though serotypes D, B, and F were reported, serotype A was also predominant [21,36,38]. The presence of the virulence gene *ptfA* demonstrated a positive association with the disease outcome of cattle, and thus, is an important epidemiological marker gene for characterizing *P. multocida* isolates [40]. In the rabbit, serogroups A, D, and F were isolated. Type A was the most frequently identified as a major cause of rhinitis and pneumonia. However, type D was detected in metritis, while type F was detected in mastitis, otitis, subcutaneous abscesses, and septicemia. The *sodC* gene was present in a single strain that was tested. Among the strains that caused respiratory lesions, the *pfhA* gene was commonly detected in type A strains compared with type D strains. Moreover, pfhA was identified in all F strains. Regarding the strains associated with rhinitis, the *fur* gene was more frequently isolated from type D strains. The *hgbB* genes were predominantly detected in the strains that were responsible for metritis [50]. The proportions of virulence genes and their associated hosts reported by various scholars are described in Figure 2.

Molecular epidemiology describes the association between virulence genes and related serotypes of *P. multocida*. There were significant associations observed between capsular serotypes and virulence genes. Accordingly, there is a relationship between capsular serogroup A and *ompH1*, *ompH3*, *plpE,* and *pfhB1*; capsular serogroup B and *hgbA* and *ptfA;* and capsular serogroup F and *ptfA* and *PlpP* [35]. Furthermore, notable links were found between *pfhA, tadD, tbpA*, and hayluniradase and the capsule type A isolates [21]. In swine and poultry cases, *toxA* and *hsf-1* showed a significant relationship with subgroup D, and *pmHAS* and *pfhA* with serogroup A [56]. The proportions of virulence genes and their associated capsular serotypes reported by various scholars are described in Figure 2.

In strains with similar genetic backgrounds, the molecular mass variability of *ompA* and *ompH* indicates that these proteins possess diversifying selection within the host. This implies they likely play a significant role in host–pathogen interactions. Comparing the outer membrane protein profile of bovine isolates with avian, ovine, and porcine species strains clearly shows that a large percentage of respiratory diseases of each species are caused by distinct strains of *P. multocida*. The discovery of closely related strains across multiple species suggests bacteria transfer between host species as part of the dynamics, structure, and interaction within *P. multocida* bacteria. However, this variability allows the bacterium to evade immune detection, colonize diverse host niches, and maintain virulence [38].

*P. multocida* encodes proteins such as those for the outer membrane and porin proteins (especially *oma87*, *psl*, and *ompH*), type 4 fimbriae (denoted by *ptfA* and *pfhA*), neuraminidase (including *nanB* and *nanH*), proteins involved in iron acquisition (*exbBD-tonB*, *tbpA*, *hgbA*, and *hgbB*), dermonecrotoxin (*toxA*), and superoxide dismutase (*sodA* and *sodC*). Among these, certain antigens have the potential to act as vaccine candidates [2,29]. Additionally, colonization factors (*ptfA*, *fimA*, and *hsf-2*), *nanH*, and outer membrane proteins are common characteristics of serogroups A and D. The circulation of these virulence gene patterns offers several indications. It suggests that certain factors involved in cross-protection could potentially be developed into vaccine prospects. These candidates may be able to generate homologs of protective immunity against all serotypes of *P. multocida* [27,28].

## 3. Immunizations and Types of Vaccine and Vaccine Candidates

### 3.1. Immunization Types

#### 3.1.1. Passive Immunization

Several passive immunizations were reported regarding hyperimmune serum and monoclonal antibodies. Passively acquired anti-*P. multocida* OMP antibodies were verified in dairy farm sera within 24 h after; the *P. multocida* antibody concentrations were found to be good until 156–419 days and then declined after 198 to 490 days of age [60]. Furthermore, serum antibody concentrations were also associated with different farm management factors [16]. Another report indicated the passive transfer of antibodies challenged with strains A: 1 (75%) and B: 2 (50%). Mice protected against a challenge using the phage lysate inactivation method suggested that the antibody alone could provide cross-protection against the serovar of a *P. mutocida* infection [61].

Six monoclonal antibodies (MAbs) against *P. multocida* serotype B:2 LPS were tested for passive protection in mice. The MAbs interacted with B:2 and serotype 5 LPSs, but not others. This mix of MAbs gave partial, serotype-specific protection, extending the mouse survival slightly. This implies that non-LPS antigens are involved in the immune response too [62].

Basic trials against swine atrophic rhinitis indicate that pregnant sows that received immunization with recombinant subunit PMT had more maternal antibodies present in their colostrum compared with those who were immunized with a conventional PAR toxoid vaccine [63]. A nontoxic truncated form of recombinant PMT (PMT2.3) vaccination of mice also develops a high level of anti-PMT antibodies with a high neutralizing serum antibody titer. High levels of serum antibody titers, cellular immunity, and good growth performance were observed in offspring from sows vaccinated with PMT2.3 [64]. Protective levels of maternal antibody were transferred effectively to progenies via the colostrum [65].

A report of two foremost outer membrane proteins of *P. multocida,* incorporating *ompH* and *ompA,* was used to continuously deliver specific MAbs to mice with hybridoma tumors, demonstrating that IgG MAbs against *ompH* are involved in the protection of mice against a lethal infection challenge using opsonization and inhibition of multiplication of *P. multocida* as a result of the increased PMN influx into the infection site [66]. Furthermore, rabbit antiserum prepared against recombinant *Oma87* passively protected mice against a homologous infection [28].

Cross-protection was observed in the oldest work on sheep immunization, indicating that a formalin-inactivated serotype D strain vaccine serum protects mice against four strains of serotype A and three nontypable strains but does not protect against types B and E [67]. Similarly, live *P. multocida*-serotypes-B:3,4-vaccinated cattle serum cross-protected passively immunized mice against *P. multocida* serotypes E:2, F:3,4, and A:3,4 [68].

#### 3.1.2. Active Immunization

Vaccination represents the most effective approach for preventing the transmission and spread of disease outbreaks by boosting the immune systems of animals against harmful pathogens. As such, it ensures comprehensive security regarding animal health and has a positive impact on public health. Given the repeated emphasis on the importance of vaccination, research focused on large-scale industrial vaccine production for the control of the disease has become crucial. This is essential to satisfy the ever-increasing demand for such vaccines [25].

Existing vaccines designed to combat *P. multocida* are of two main types: inactivated whole-cell bacterin vaccines and live attenuated bacterial vaccines. Unfortunately, both of these vaccine varieties have drawbacks. They are known for having inconsistent safety records, being less effective, and having a relatively short-lived protective period. In particular, bacterin vaccines typically safeguard against the disease for less than six months, while live attenuated vaccines offer prevention for around a year [69]. The weak efficacy of existing vaccines might be attributed to several factors. Poor immunogenicity (poor at triggering cellular immunity) could be one reason, as well as reversion to virulence of live attenuated strains. There is also an issue of vaccine mismatch between the vaccinal serotype strain and the circulating wild-type bacteria, which would prevent effective targeting. Additionally, these vaccines often fail to induce long-lasting immunity. Safety is another major concern, especially with bacterins. The high dose of bacterial-derived endotoxin and other substances in the administered doses of bacterins can lead to systematic reactions [26].

However, one must recognize that inactivated vaccines, even though they produce short-lived immunity and offer insufficient cross-protection, are effective and possess an encouraging cost–benefit ratio [70]. Furthermore, compared with other vaccine types, inactivated vaccines face fewer regulatory restrictions. Even with the progress in genetic engineering and biotechnology, in the short term (three to five years), inactivated vaccines and those adjuvanted with aluminum hydroxide will sustain their commercial dominance. The future of bacterial vaccines in animal production looks bright. Thanks to improvements in vaccine formulations and bioengineering, these vaccines hold the potential to enhance the sustainability of the vaccine production industry. However, it is essential to research methods to boost the efficacy of vaccines and increase their accessibility [71].

The commonly used formalin- or heat-killed whole-cell bacteria vaccines are also commonly circulating vaccines that may affect some major epitopes of Pasteurella. Oil-adjuvanted formulation and/or saponin-added vaccine preparations showed variable efficiency in long-term immunity for cattle, chickens, ducks, turkeys, and rabbits, but not sheep. Alum, Freund’s incomplete adjuvant, PMT toxoid, and DNA were all utilized concurrently to augment the impact of bacteria. Typically, the inactivated bacterin was unable to provide cross-protection or long-lasting immunity. Moreover, it induces local inflammation at the site of administration [24].

In summary, most works on vaccine developments related to virus-like particles (VLPs) hold greater possibilities as a vaccine platform. Their distinguishing characteristics enable them to boost the immune response, and they can serve as a vehicle for foreign antigens to combat infectious diseases. Vaccines formulated based on VLPs are part of the new-generation vaccine strategies that have received approval. Most studies on these vaccines were in the advanced stage of evaluation [70]. Most available vaccines and vaccine candidates for their target host, antigenic composition, method of delivery, and drawbacks summarized below in Table 2, Table 3 and Table 4.

### 3.2. Vaccine Types

#### 3.2.1. Whole-Cell Vaccine

As previously discussed, administering vaccinations using either inactivated bacterin or live attenuated bacteria represents an efficient and cost-effective approach. This method serves to enhance the health status of animals and safeguard them from *P. multocida* infections [24]. However, bacterins against *P. multocida* offer restricted protection against heterologous serotypes. Moreover, there is a concern that live/attenuated vaccines might revert to a virulent state, which could then lead to the infection of animals [72].

These currently available biological products, such as modified live vaccines and bactrins, have their potencies tested based on counting bacterial colonies. Regarding the bacterin potency, the approach involves vaccinating and then challenging mice and/or birds. Inactivated vaccines are used for the prevention of atrophic rhinitis. However, there are no standard criteria for potency testing of *P. multocida* type D toxoid. Somatic antigens, especially lipopolysaccharide (LPS), seem to play a potential role in the immune mechanism [73].

The OIE [9] recommends that the effective vaccines against hemorrhagic septicemia are formalin-killed bacterins or dense bacterins with an adjuvant. These adjuvant-added vaccines not only increase the level of immunity but also extend its duration. The seed culture used in vaccine production must consist of a bacterial capsule. Vaccines are standardized according to their bacterial load, which is determined through turbidity tests and measurement of the dry bacterial weight. Potency tests are typically conducted using rodents.

Currently, bovine vaccines for *P. multocida* disease are available on the market; however, these vaccines are mainly restricted to aluminum-adjuvanted whole killed bacterins [71] or live attenuated vaccines [69], both of which provide protection specific to certain serogroups. However, they can cause strong reactions and are not well-matched to the currently circulating strains. This situation makes it extremely challenging to standardize the vaccines, carry out large-scale production, and maintain quality control during vaccine manufacturing. It is important to reduce the drawbacks of traditional vaccine development of modified live attenuated vaccines, such as PMZ2 for ducks [74] and PmCQ2Δ4555–4580 for cattle [75]; these vaccines exhibit cross-protection, yet their safety needs further verification. Even though there are several commercially accessible vaccine formulations, such as alum-precipitated, oil-adjuvanted, and multiple-emulsion vaccines, the pursuit of appropriate, highly protective hemorrhagic septicemia vaccines that offer long-lasting immunity is growing in intensity. Simultaneously, efforts are underway to clarify uncertainties regarding the bacteria, including their virulence factor, pathogenesis, immune mechanism, and diversity within *P. multocida* organisms [72].

Whole-cell bacterins can offer a certain level of protection. However, this protection is restricted to the homologous lipopolysaccharide (LPS) serotypes. There is substantial information indicating that cross-protecting antigens are expressed only in in vivo situations. Live attenuated vaccines developed empirically have the potential to safeguard against heterologous serotypes. Nevertheless, since the attenuation mechanism is not clearly defined, the reversion of these vaccines to virulence is quite common [28].

The OIE [76] recommends several commercial vaccines for atrophic rhinitis control. These include whole-cell bacterins *B. bronchiseptica* combined with toxigenic *P. multocida* bacterin (capsular type D) and/or a *P.multocida* toxoid. Moreover, some toxigenic and non-toxigenic type A strains of live attenuated *B. bronchiseptica* vaccines exist too. Vaccines prepared with only *B. bronchiseptica* are unsuitable for PAR, except in non-progressive cases. *P. multocida* and *B. bronchiseptica* vaccines reduce bacterial colonization but do not eliminate them or stop infection. Most commercial vaccines have an oil adjuvant or aluminum hydroxide gel.

#### 3.2.2. Killed Vaccines

According to the OIE [9], there are three vaccines for preventing hemorrhagic septicemia (HS): whole-cell bacterins, alum-precipitated vaccine (APV), and oil-adjuvant vaccine (OAV). Bacterins require frequent vaccination for adequate immunity and can cause shock reactions when dense, while the APV has fewer shock reactions, and the OAV has virtually none.

A formalin-killed vaccine strain of type D4 shows significant protection against homologs of six type D strains and heterologous strains of four type A strains [67]. This whole-broth-culture formalin-killed *P. multocida* type was widely used for sheep and goats in Ethiopia, NVI [77], but some improvements were made to the killed *P. multocida* type A antigens formulated with bacterial DNA as an adjuvant as a new vaccine against *P. multocida* in sheep [78]. However, the vaccine applied against *P. multocida* bio-type A was found to be less effective in developing protective antibodies against sheep disease; this may have been because the circulating *P. multocida* serotypes may have been present in low amounts with the vaccine applied. Accordingly, the creation of a multivalent vaccine that incorporates the most commonly occurring Pasteurella serotypes sourced from diverse geographical origins is expected to facilitate efficient prevention. This requires further study on the identification of strains of *P. multocida* in sheep for further multivalent vaccine development [79].

The initial vaccine for HS was developed in the early 1900s. Unfortunately, the bacterin vaccine elicited weak antibody responses. It provided immunity for merely six months and induced a certain degree of shock in animals. To address the protein shock issue associated with killed-broth-culture vaccine organisms, formalized or agar-wash heat-killed vaccines were developed. These improved vaccines offered immunity for up to four months [72]. Some of the main types of killed vaccines include bacterins, alum-precipitated vaccines, aluminum hydroxide gel vaccines, oil-adjuvanted vaccines, and multiple-emulsion vaccines. Bacterin represents the most basic form of an HS vaccine. It is produced from killed *P. multocida* using either physical methods, such as heat, drying, or ultraviolet radiation, or chemical agents, such as phenol, Lysol, formalin, or sodium azide [69].

In swine, an inactivated whole-cell antigen of multiple serogroups A (L3 and L6) and D (L6) aluminum hydroxide gel adjuvant vaccines produced no heterologous protection of type A (L3 to L6), but some cross-protection was absorbed from serotype D6 against heterologous strains [80]. In chickens, a formalin inactivated with alum adjuvant vaccine of serotype A was commonly used but improvements were made through *P. multocida* A:1 grown in the presence of low FeCl_3_ concentrations, inactivated with a high FeCl_3_ concentration, and adjuvanted with bacterial DNA from *P. multocida* type B:2 containing immune-stimulatory CPG motifs that protect chickens with a lethal *P. multocida* A:1 [81,82].

#### 3.2.3. Live Attenuated Vaccine

This vaccine formulation strategy involving attenuated or avirulent vaccines can be implemented through various methods, including subjecting the bacteria to an iron-deficient condition, using chemically mutagenic substances, and deleting virulence genes [26]. A *P. multocida* strain B:3,4, specifically a fallow deer strain, was used to prepare a live HS vaccine that has now been utilized for disease control in cattle and water buffalo older than six months, administered as an intranasal aerosol application [68]. Serum of vaccinated cattle cross-protected mice against infection with the serotypes E:2; F:3,4 and A:3,4. The OIE [9] reported that the Food and Agriculture Organization (FAO) has recommended the vaccine as safe and effective for use in Asian countries. Nevertheless, there are no reports of utilization in other countries. In countries affected by hemorrhagic septicemia (HS), only killed vaccines are currently in use. Similar reports also discuss a type A mutant strain, *PmCQ2Δ4555*, *4580*, that was a wild-type strain *PmCQ2* with six obvious genes missing; it can protect mice challenged by serogroup B and slightly protect against serogroup F [76].

In an older study by Rice et al. [83], which built on the aforementioned work, the vaccination routes for chickens using a live avirulent *P. multocida* vaccine were evaluated. Across every experiment, the subcutaneous vaccination route offered the most substantial protection. Among broilers vaccinated via the subcutaneous route, the protection levels scored 95% and 97.5%. Notably, there were no occurrences of unwanted lesions or cheesy mass formation beneath the skin on the backs of broilers’ necks. A novel, live attenuated *P. multocida* vaccine strain for ducks named PMZ2 features a deletion of the *gatA* gene and the initial four bases of the *hptE* gene. These genes play crucial roles in the synthesis of the lipopolysaccharide (LPS) outer core. Despite its cross-protection capabilities remaining unstudied, PMZ2 is a promising live attenuated vaccine for ducks, with the potential for delivery via oral and intranasal routes [74].

#### 3.2.4. Recombinant Subunit Vaccine

Subunit vaccines enable straightforward large-scale production. Additionally, they possess the capacity to modify and enhance proteins. Subunit vaccines comprise individual immunogenic bacterial components, such as proteins and polysaccharides, that can confer immunity [24,63]. In this bacterium, most genes were found to encode membrane proteins through a bioinformatics analysis of the *P. multicida* genome, where an extensive set of OMPs and outer membrane-associated portions were identified. Given their predicted localization as either secreted or surface-exposed proteins, these proteins were examined for their immunogenicity and capacity to safeguard against lethal *P. multocida* infections. However, only the recombinant protein plpE was found to be more likely to trigger a protective immune response [30].

An in silico analysis revealed that unique B and T cell epitopes determined based on the adopted antigenic LPS outer membrane complex proteins found in fowl, buffalo, and goats can be appropriate targets for vaccine development against fowl cholera (FC) and hemorrhagic septicemia (HS) [84]. Similarly, Mostaan et al. [85] reported on the immunogenicity, antigenicity, various serotype coverage, half-life, and antibody accessibility epitopes of PlpE (regions 1 + 2 + 3), including regions of the *P. multocida* plpE protein that can be used as an appropriate serotype-independent vaccine candidate against pasteurellosis.

In bovine pasteurellosis, part of the bacteria’s surface lipoprotein (*PmSLP-3)* formulated with Montanide ISA 61 fully protects against a serogroup B challenge by creating sustained serum IgG titers that stay for 3 years after administration of two doses and can simultaneously protect against a serogroup E challenge [86]. Similarly, vaccines comprising *PmSLP* (1 and 3) antigens can be applied as an effective solution for preventing HS- and BRD-related *P. multocida* infections, indicating that *PMSLP* is an important vaccine component [87]. *PlpE* has been reported as a crucial cross-reactive outer membrane protein in *P. multocida*. The plpE genes of *P. multocida* serotypes A:3, B:2, and D1 were studied, followed by the expression and immunoblotting analysis of *plpE* from B:2. OMPs are powerful immunogens that offer protection to mice, rabbits, chickens, and calves. *P. multocida* A:3 (strain P1059) was discovered to be cross-protective, inducing both active and passive cross-protection in chickens and turkeys. It was also found that the *plpE* gene of hemorrhagic septicemia which causes serotype B:2, expressed in *E. coli,* and the recombinant *plpE* was strongly immunostained by antiserum against the whole-cell antigen. This indicates that the protein is expressed in vivo [88]. Similarly, Wu et al. [89] cloned *P. multocida* lipoprotein E (*PlpE)* from *P. multocida* strain X-73 (A:1) and expressed it in Escherichia coli to show that mice and chickens immunized with *r-PlpE* were protected against challenge infections with serotypes A:1, A:3, and A:4. Therefore, *plpE* serves as a cross-protective antigen.

According to Okay et al. [90], recombinant *OmpH*, *plpE*, and *plpE-OmpH* fusion protein formulated with oil-based CpG oligodeoxy nucleotide stimulated 100% protection, indicating that the recombinant *PlpE* is a possible acellular vaccine candidate for cattle. In avian species, new vaccine formulations involving serotype A:1 recombinant *VacJ, PlpE,* and *ompH* [91] and A:3 recombinant adhesive protein (rCp39) [92] for duck and chicken, respectively, are also very promising recombinant vaccine candidates.

In regard to swine, *P. multocida* toxin (PMT2.3) vaccination in mice delivered a high anti-PMT antibody level with a high neutralizing antibody titer and cellular immune response, with high levels of serum antibody titers and growth performance passed down to their offspring. Therefore, PMT2.3 in the truncated and nontoxigenic recombinant PMT form is a good candidate subunit vaccine against PAR-induced infection in pigs [64]. Similarly, recombinant subunit *P. multocida* toxin (*rsPMT*) containing either the N or C terminal portion of PMT developed high neutralizing antibody titers [63]. Wang et al. [93] found that a recombinant vaccination of *rTorA, rPrx,* and/or *rPGAM* of serogroup D proteins also protected 60~80% of the tested mice against the challenge with *P. multocida* field isolates of A, B, and F, and thus, was found to be a good vaccine prospect.

#### 3.2.5. DNA Vaccine

DNA vaccines have been put forward as a possible solution. As subunit vaccines, they carry no infective component or reversion to a virulence state. These vaccines have the advantage of enabling simultaneous immunization against numerous pathogens. They are comparatively straightforward to formulate and cost-effective to produce. The administration of DNA vaccines has been demonstrated to trigger immune responses and offer protection against trials in various animal models. However, despite these benefits, they are not currently suitable for mass vaccination. This is because their application requires biological or physical carriers [94]. Further shortcomings, such as the potential to impact related cell growth, the risk of prompting the production of antibodies against DNA, the development of tolerance to the antigen (protein) generated, and the possibility of abnormal processing of bacterial protein, are factors that can be seen as limiting the development of these types of vaccine [85].

A DNA vaccine constructed using two distinct outer membrane proteins, namely, *pOMPH* and *pOMPA*, demonstrated its ability to safeguard against avian pasteurellosis. Moreover, the fusion vaccine developed from these two OMPs exhibited the highest level of protection [95]. An *ompH* gene amplified from *P. multocida* strains was sequenced, found one restriction site, found two fragments. The *OmpH* gene DNA expressed in *E. coli* protects vaccinated rats, which makes it a candidate vaccine for major farm animals [96].

*OmpH* and *ompA* are two major immunogenic proteins in avian *P. multocida.* Studies on the DNA genes of these OMPs found that a divalent combination (*pcDNA-OMPH + pcDNA-OMPA, pOMPH + pOMPA*) and a combination of the two gene vaccines (*pcDNA-OMPH/OMPA, pOMPHA*) provide good protection, equivalent to the attenuated live vaccine of fowl cholera, while the monovalent form is not protective unless combined [97]. Again, chitosan showed its potential to trigger the immune response associated with a naked DNA vaccine centered on *ptfA* of *P. multocida*. Thus, the *ptfA* chitosan construct provides robust protection against *P. multocida* [98].

Conventional vaccines, such as alum-precipitated and oil-adjuvanted broth bacteria, were subcutaneously injected to protect against hemorrhagic septicemia. Unfortunately, this offered only short-duration immunity and demanded frequent administration. Gene fragments from *P. multocida* serotype B (ABA39) were sub-cloned into DNA expression plasmid pVAX1-ABA39; research work showed the resulting recombinant vaccine has the potential to be a successful future vaccine against HS [99].

**Table 2 vaccines-13-01034-t002:** Cattle, buffalo, sheep, and goat *P. multocida* vaccines and vaccine candidates.

Target Host/Animal Species	Type of Vaccine	Target Serotype and Strain to Protect	Antigenic Composition	Administration Route	Immunological Effect	Drawbacks	Animal Immunized and Status	References
Bovine	Live attenuated	B:2	Wild-type strain 85020 contains a deleted *aroA* gene (JRMT12)	Intramuscular (IM)	Higher IgG and IgM	Dose-dependent	10^8^ CFU was safe and effective on calves	[100]
Bovine	Live attenuated	A	PmCQ2Δ4555–4580 wild-type strain PmCQ2, with six obvious genes missing	IM	100% protection against A and B, 40% against F, good cross-protection against B, and slightly protects against F		Trial with mice	[75]
Cattle and buffalo HS	Killed	B6	B6	Subcutaneous (SC)	Specific but 100% protection	4 to 6 months immunity	Cattle and buffalo in use	[69]
Cattle and buffalo	Live	B2	B3,4	SC	9 to 12 months of protection	Route of administration and serotype mismatch	Cattle and buffalo in use	[69]
Cattle	Formalin-killed	B2	B2	SC	6 to 8 months of protection	Anaphylactic shock in some animals	Cattle in use	[77]
Best with cattle/buffalo	Live	B2	B3,4	Intranasal	High AB titer; E:2, F:3,4 and A:3,4	-	Cattle and buffalo in use	[68]
Cattle	Live	*P. multocida* A:3	A:3	IM	Reduced clinical lesions		Cattle	[101]
Bovine	Recombinant	*P. multocida* A:3	Recombinant proteins *PlpE* and *PlpEC-OmpH*	Intraperitoneal	100% protection; increased IgG and serum IFN-gamma	-	Mice trial	[90]
Bovine and buffalo	Subunit	B2	Native OMP	Subcutaneous	100% protection	-	Mice	[102]
Cattle	Subunit	B and E	PmSLP-3	SC/IM	Highest level of mucosal *PmSLP-3* specific IgG; cross-protection with serogroup E	No cross-protection against BRD strains	Cattle and mice	[86]
Cattle with BRD/fowl cholera	Recombinant subunit	A (*P. multocida* P488 challenge)	OMVs (*OmpA*, *OmpH*, and P6)	Intranasal	High AB titer and mucosal immune responses, cross-protection with M, and hemolytic		Mice trial	[4]
Cattle HS/BRD	Bivalent subunit	B2 and A3	PmSLP-1 (BRD-PmSLP) and PmSLP-3 (HS-PmSLP)	SC/IM	High serum IgG and a good vaccine with good cross-protection		Cattle	[87]
Cattle	Subunit	B:2	B:2 OMPs plus anti-idiotype AB	SC	100% protection in rabbits, better protection than whole bacterin	-	Rabbit trial	[103]
Bovine	Recombinant subunit	A, B, F	OMPs of A, B, and F	SC	Effective against A and B	Still needs verification in cattle	Mice	[104]
Cattle HS	Recombinant	B:2	r*OmpH* adjuvanted with CpG-ODN	Intranasal	High serum IgG and secretory IgA levels		Calves	[105]
Cattle and buffalo	DNA	B	B2: Clone pVAX1- ABA392	IM	Increased serum IgG and no organ lesions		Rat trial	[99]
To all but cattle strain as a challenge	DNA	Mainly A	*The ompH* conserved gene of ten strains	IM	High AB titer with good cross-protection		Mice trial	[96]
Cattle and buffalo	DNA	B:2	*tbpA* gene of B:2	-	Increases humoral and cell-mediated immune response	-	Mice trial	[106]
Goat	Inactivated recombinant	B:2	B:2 fimbrial protein	Intranasal	High IgG and IgA		Goat trial	[107]
Sheep	Inactivated multivalent	A and D4	D4 and type A (8473 strain)	SC	Good cross-protection against some A strains	The short duration of immunity (6 months)	Sheep	[67]
Sheep	Inactivated/killed	A (challenge strain PMSHI-9)	A+ iron inactivation with iron and bDNA adjuvant type A	SC	Higher Ab titer and cellular (IL-4, IFN-γ, and TNF-α); good humoral and cellular immunity and safety	-	Sheep	[78]
Sheep and goat	Formalin-killed	A	A	SC	6 to 8 months of protection	-	Sheep and goats in use	[77]

**Table 3 vaccines-13-01034-t003:** Poultry and avian species related to *P. multocida* vaccines and vaccine candidates.

Target Host/Animal Species	Type of Vaccine	Target Serotype and Strain to Protect	Antigenic Composition	Administration Route	Immunological Effect	Drawbacks	Animal Immunized and Status	References
Duck	Live attenuated	A LPS1/PMZ2	PMZ2 gene with the deleted *gatA* gene and part of the *hptE* gene	Oral and intranasal	High level of serum IgG with strong bactericidal effect and a significant increase in IgA response	Cross-protection has not evaluated fully	Duck trial with good effect	[74]
Chicken	DNA	A1 challenge study	*ptfA* gene with added chitosan particle	IM	High AB concentration	Only a 68% protection level	Chicken trial	[98]
Chicken	DNA	*P. multocida* CVCC474 strain challenge	*ompA* and *ompH*	IM	High AB and equivalent protection against the attenuated live vaccine	The delivery system is not appropriate	Chicken trial	[97]
Turkey	Recombinant peptide	*P. multocida* x73 (A:1) challenge	A:3 FHAB2 peptides (filament)	Intradermal	Reduce mortality and organ pathology with cross-protection		Turkey trial	[108]
Duck	Recombinant subunit	A:1	A:1 (PMWSG-4) recombinant *VacJ, PlpE,* and *ompH*	SC	100% protection; reduces tissue damage and colonization	-	Duck	[91]
Chicken	Inactivated	A1	Inactivated A1 + B2 DNA adjuvant	SC	Cost reduction with safe and innate stimulation	-	Chicken	[81]
Chicken	Recombinant PlpE	A1	Recombinant *PlpE* of (A:1)	SC	80–100% survival rate with cross-protection (A:3,4)	-	Mice trial	[89]
Chicken	Live	A and B	A virulent (A1)	SC	95 to 97.5% protection level	-	Chicken	[83]
Chicken	Inactivated with formalin and FeCl3	A	A:1 adjuvant with bDNA	SC	100% long-term protection and good humoral and cellular immunity in mice	-	Mice trial	[82]
Chicken	Recombinant protein	A1	A:3 recombinant adhesive protein (rCp39)	SC	Cross-protection and 60 to 100% protection		Chicken	[92]
Chicken	DNA	A	A p*OmpH* and p*OmpA*	IM	Higher AB titer than the live attenuated vaccine	The vaccine delivery system is not appropriate	Chicken	[95]
Duck	Recombinant	A:1	rHVT (herpes virus) *OmpH*	IM	Ensures good safety and protection	Duration may be short	Duck	[109]
Chicken	Recombinant	A:1	A (r*OmpH*)	IM	100% protection from fowl cholera		Chicken	[110]

**Table 4 vaccines-13-01034-t004:** Swine- and rabbit-related *P. multocida* vaccines and vaccine candidates.

Target Host/Animal Species	Type of Vaccine	Target Serotype and Strain to Protect	Antigenic Composition of Vaccine	Administration Route	Immunological Effect	Drawbacks	Animal Immunized and Status	References
Swine	DNA	Wild-type *P. multocida* strain 4533	*toxA*	IM	Secrete IFN_-ᵞ,_ increased AB titer	Toxin-specific only	Swine	[111]
Rabbit	Killed	A and F	*pfhA*, *sodC*, *soda*, *exbB*, *oma87*, *fur*, *fim4*, *nanB*, *nanH*, and *fimA*	--	Reduced the severity of the disease	Still does not protect against some strains of A and D	Rabbit	[112]
Swine	Killed	A and D	A:L3 D:L6	SC	High serum IgG	No cross-protection against others	Mice trial	[80]
Pig	Recombinant subunit	PMT (toxA)	Recombinant *toxA (Tox1*, *Tox2*, and *Tox7)*	IM	Protective humoral and cellular immunity	-	Pig	[63]
Rabbit	Subunit	A3	A:3 OMP (IROMP) adjuvanted with cholera toxin (CT)	Intranasal	Mucosal and systemic AB increased; reduced bacterial count	Only reduced bacterial count	Rabbit	[113]
Swine	Recombinant subunit and whole-cell bacterin	A and D	D (*toxA*) and D and A	IM	Controlled prevalence and severity of the disease		Piglet	[65]
Swine	Recombinant subunit	D	*toxA* (PMT2, 3)	SC	Good humoral and cellular immune responses with passive transfer		Swine	[64]
Swine	Recombinant	A and D	Full-length rOmpH	SC	High Ab		Mice	[114]
Swine	Recombinant subunit	A	Serogroup D (rTorA, rPrx, and/or rPGAM)	Intraperitoneal	High antibodies and IFN-γ, IL-4, and IL-10 in mice with good cross-protection		Mice trial	[93]
Swine	Recombinant bivalent subunit	A	PMT NC adjuvanted with rSly or CpG	IM	Enhanced humoraand cellular immune responses		Piglet	[115]

## 4. Conclusions and Future Directions

*Pasteurella multocida* is an economically important livestock disease that causes various diseases in animals and opportunistic infections in humans. It has four subspecies, five capsular serogroups, and 16 somatic serotypes. Virulence factors include capsular polysaccharides, LPSs; LPS, PMT, and iron acquisition genes; and OMP relationships responsible for phagocytosis, inflammation, adhesion, and nutrient acquisition. Currently, some commonly used antibiotics are developing resistance. Vaccines (inactivated and live) have limitations. Live attenuated vaccines can stimulate cross-protection, while killed vaccines confer serotype-specific immunity. Most vaccines are inactivated whole-cell bacterin vaccines and live attenuated bacterial vaccines. These vaccines have many drawbacks, such as having inconsistent safety records, being less effective, a relatively short-lived protective period, reversion to virulence of live attenuated strains, and a mismatch between the vaccinal serotype strain and circulating wild-type bacteria. Adjuvant formulations vary in immunity duration. Recombinant subunit vaccines are composed of immunogenic proteins selected through in silico analysis and bioinformatics; these are mostly tested extensively to trigger a protective immune response and can be more scalable and intended for large-scale production. Therefore, current vaccine development focuses on identifying immunogenic proteins. For example, PmSLP-3, OmpH, PlpE, and VacJ have shown immunogenic and protective properties in different animal models. In avian, swine, and other species, recombinant and subunit vaccines using these proteins show promise but need more research on antigenic formulations, adjuvants, dosages, and vaccination schedules. DNA vaccines have limited adoption but show some protection. Research on recombinant and subunit vaccines for *P. multocida* in small ruminants should be emphasized as an alternative to killed vaccines.

## Figures and Tables

**Figure 1 vaccines-13-01034-f001:**
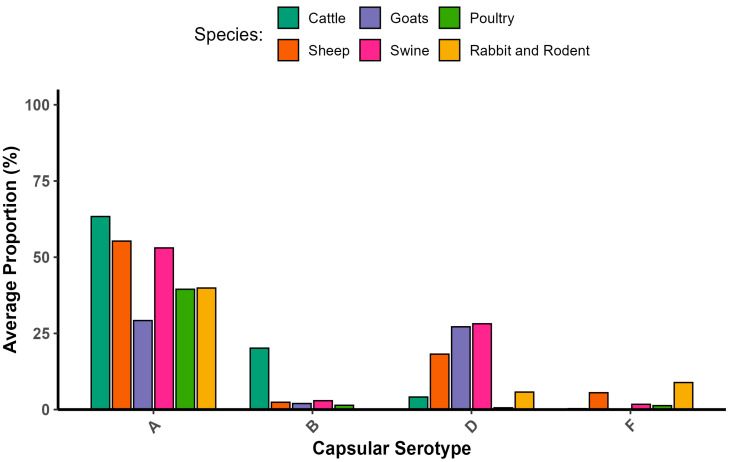
Frequency of isolation capsular serotype versus host. [2,12,20,21,22,23,34,35,36,37,38,39,40,41,42,43,44,45,46,47,48,49,50].

**Figure 2 vaccines-13-01034-f002:**
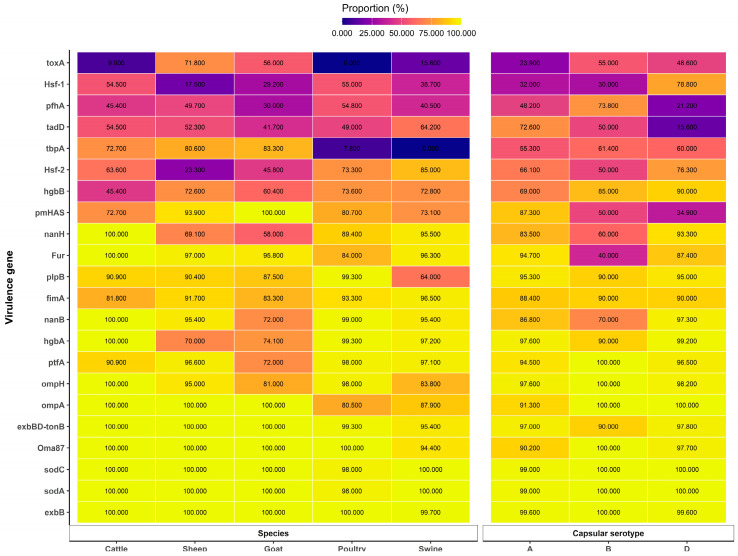
Faceted heatmap of virulence—gene proportions across host species and capsular serotypes. Each tile displays the mean percentage of isolates positive for a given virulence gene within a species group (Cattle, Sheep, Goat, Poultry, and Swine) or within a capsular serotype (A, B, and D). Values were computed by expanding multi-valued cells, parsing numeric content, excluding missing entries, averaging within each gene–category cell, and constraining results to the valid range of 0.000–100.000. Virulence genes are ordered by their overall mean prevalence across both facets to aid comparability. Sources for Figure 2: [12,22,34,36,46,47,48,49,58,59]. Notes: (ref. [12]) reported only for sheep; (refs. [47,59]) reported only for swine; (ref. [22]) reported only for swine and poultry of serogroups A and D; (ref. [49]) reported only for sheep and goats; (ref. [48]) reported only for goats; (ref. [58]) reported only for avian species and sheep.

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
