# Peer review of "Advances in Pasteurella multocida Vaccine Development: From Conventional to Next-Generation Strategies"

_vaccines, 2025, doi:10.3390/vaccines13101034_

Round 1
Reviewer 1 Report
Comments and Suggestions for Authors
The manuscript entitled “Advances in Pasteurella multocida Vaccine Development: From Conventional to Next-Generation Strategies” is a comprehensive and well-organized review that provides a timely overview of vaccine strategies against P. multocida, a major veterinary pathogen. The authors cover a broad range of platforms—from traditional bacterins to modern subunit, DNA, and OMV-based vaccines—and effectively highlight key antigens, adjuvants, and host-specific challenges. The inclusion of summary tables enhances the utility of the manuscript for researchers and practitioners.
Minor remarks:
1. In my opinion, discussion of issues - such as lack of cross-protection, short duration of immunity, and serotype diversity - would benefit from deeper analysis. For example:
There is limited discussion on why killed vaccines fail to induce robust cellular immunity or how LPS variability undermines protection.
The risk of reversion in live-attenuated vaccines and the adjuvant dependency of subunit vaccines are underemphasized, despite their critical impact on safety and field applicability.
2. The manuscript is generally well-written, but a few minor linguistic and typographical issues should be addressed to improve clarity and readability
Lines 349–350: Repetition of "oil-adjuvant vaccine (OAV)" appears to be redundant. The OIE [9] lists three types—bacterins, APV, and OAV—but the sentence structure is unclear and should be revised for accuracy.
Line 382: "Adjuvate with bacterial DNA" — "adjuvate" should be lowercase ("adjuvated") and is better phrased as "adjuvanted with".
Author Response
Comments 1: In my opinion, discussion of issues - such as lack of cross-protection, short duration of immunity, and serotype diversity - would benefit from deeper analysis. For example:
There is limited discussion on why killed vaccines fail to induce robust cellular immunity or how LPS variability undermines protection.
The risk of reversion in live-attenuated vaccines and the adjuvant dependency of subunit vaccines are underemphasized, despite their critical impact on safety and field applicability.
Response 1: Thank you, very interesting point you raised. This part required details which is not mentioned here. The reasons can be that it needs detailed writing on the immunological concept we tried to mention grossly(poor immunogenicity) from lines 252 to 262. But we get into the details mentioned there. The possible reasons for this can be: Killed vaccines were poor at triggering cellular immunity, cannot replicate, and disable MHC1 presentation. Harish's inactivation method denatures T cell epitopes as well as suboptimal activation of pattern recognition receptors. P. multocida has high LPS diversity. LPS structure is prone to phase variation and mutation. A killed vaccine can be prepared for a single bacterial strain and a few LPS types.
Comment: 2. The manuscript is generally well-written, but a few minor linguistic and typographical issues should be addressed to improve clarity and readability
Lines 349–350: Repetition of "oil-adjuvant vaccine (OAV)" appears to be redundant. The OIE [9] lists three types—bacterins, APV, and OAV—but the sentence structure is unclear and should be revised for accuracy.
Line 382: "Adjuvate with bacterial DNA" — "adjuvate" should be lowercase ("adjuvanted") and is better phrased as "adjuvanted with".
Response 2: Great to see this editorial comment. The manuscript is now already revised by the English revision of the MDPI vaccine author service. line 349-350 and line 382 revised. You can see the Changes we made on the attached, but this is before English revision.
You can refer to the latest manuscript for the overall revisions made to the manuscript after the language editorial section was made. Right now few improvements have also been made, indicated by track changes attached below
Thank you again for your constructive comments you give to us
Reviewer 2 Report
Comments and Suggestions for Authors
The authors provide an extensive review of Pasteurella species as a cause of economically and clinically significant infections in livestock, other animals, and humans. They also provide an extensive survey of vaccination approaches that have been used to combat this infectious agent, including types of vaccines and targeted antigens. They summarize results on efficacy of these vaccines and identify limitations and drawbacks. The literature review seems reasonably thorough, and the manuscript provides a useful reference list to the primary literature.
The main problem is the writing. The manuscript is dense, poorly organized, and displays frequent instances of very unclear writing, sometimes to the degree of incomprehensible gibberish. A non-exhaustive list of examples is provided below. Ultimately the authors provide a useful citation list, but largely fail in constructing a useful review amalgamating and summarizing the current literature.
Line 100: “adhesions” should be “adhesins”
Lines 104-106: “Suggested that it will be essential to develop other combination strategies to produce vaccines against pathogens that present several alternative iron-uptake proteins . ” This is not a sentence.
Lines 116-118: “Whereas type B contains a polysaccharide capsule consisting of galactose, arabinose, and mannose sugar residue), and type F chondroitin or type D heparin/heparin surface. ” This is not a sentence.
Lines 162-164: “The presence of the virulence gene ptfA shows that a positive association with the disease outcome of cattle found that a positive association with the disease outcome in cattle, …” This is redundant and unclear.
Lines 189-191: “The discovery of closely related strains across multiple species suggests transfer of bacteria between host species is considered in the population biology of P.multocida.” This is not a sentence and the meaning is unclear.
Lines 350-351: “... whole cell bacterins, alum precipitated vaccine (APV), and oil - adjuvant vaccine (OAV), and oil adjuvanted vaccine (OAV).” This is redundant.
Lines 429-434: “In bovine pasteurellosis part of the bacteria surface lipoprotein (PmSLP-3) formulated with Montanide ISA 61 fully protective for serogroup B challenge create sustained serum IgG titers that stay for 3 years after administration of two doses and simultaneously can protect serogroup E challenge, similarly vaccines composed of PmSLP (1 and 3) antigens can be a applied and effective solution for preventing HS and BRD related P. multocida infections.” This is largely incomprehensible gibberish.
Line 505: “Antibiotics are ineffective.” This is at least misleading, or untrue to a considerable degree. Antibiotic therapy is often effective if the diagnosis is considered and determined quickly, but clearly has limitations that make vaccination a useful complementary approach.
Comments on the Quality of English LanguageThe authors provide an extensive review of Pasteurella species as a cause of economically and clinically significant infections in livestock, other animals, and humans. They also provide an extensive survey of vaccination approaches that have been used to combat this infectious agent, including types of vaccines and targeted antigens. They summarize results on efficacy of these vaccines and identify limitations and drawbacks. The literature review seems reasonably thorough, and the manuscript provides a useful reference list to the primary literature.
The main problem is the writing. The manuscript is dense, poorly organized, and displays frequent instances of very unclear writing, sometimes to the degree of incomprehensible gibberish. A non-exhaustive list of examples is provided below. Ultimately the authors provide a useful citation list, but largely fail in constructing a useful review amalgamating and summarizing the current literature.
Line 100: “adhesions” should be “adhesins”
Lines 104-106: “Suggested that it will be essential to develop other combination strategies to produce vaccines against pathogens that present several alternative iron-uptake proteins . ” This is not a sentence.
Lines 116-118: “Whereas type B contains a polysaccharide capsule consisting of galactose, arabinose, and mannose sugar residue), and type F chondroitin or type D heparin/heparin surface. ” This is not a sentence.
Lines 162-164: “The presence of the virulence gene ptfA shows that a positive association with the disease outcome of cattle found that a positive association with the disease outcome in cattle, …” This is redundant and unclear.
Lines 189-191: “The discovery of closely related strains across multiple species suggests transfer of bacteria between host species is considered in the population biology of P.multocida.” This is not a sentence and the meaning is unclear.
Lines 350-351: “... whole cell bacterins, alum precipitated vaccine (APV), and oil - adjuvant vaccine (OAV), and oil adjuvanted vaccine (OAV).” This is redundant.
Lines 429-434: “In bovine pasteurellosis part of the bacteria surface lipoprotein (PmSLP-3) formulated with Montanide ISA 61 fully protective for serogroup B challenge create sustained serum 430 IgG titers that stay for 3 years after administration of two doses and simultaneously can protect serogroup E challenge [86], similarly vaccines composed of PmSLP (1 and 3) antigens can be a applied and effective solution for preventing HS and BRD related P. multocida infections.” This is largely incomprehensible gibberish.
Line 505: “Antibiotics are ineffective.” This is at least misleading, or untrue to a considerable degree. Antibiotic therapy is often effective if the diagnosis is considered and determined quickly, but clearly has limitations that make vaccination a useful complementary approach.
Author Response
Comments 1: The authors provide an extensive review of Pasteurella species as a cause of economically and clinically significant infections in livestock, other animals, and humans. They also provide an extensive survey of vaccination approaches that have been used to combat this infectious agent, including types of vaccines and targeted antigens. They summarize results on efficacy of these vaccines and identify limitations and drawbacks. The literature review seems reasonably thorough, and the manuscript provides a useful reference list to the primary literature.
The main problem is the writing. The manuscript is dense, poorly organized, and displays frequent instances of very unclear writing, sometimes to the degree of incomprehensible gibberish. A non-exhaustive list of examples is provided below. Ultimately the authors provide a useful citation list, but largely fail in constructing a useful review amalgamating and summarizing the current literature.
Response 1: yes look it very well, now we already made English revision by the MDPI Vaccine author service. You can see the final clean version attachment. the Editorial comments you give us you can see the revised version of it labeled in red marks
Comments 2:
Line 100: “adhesions” should be “adhesins”
Lines 104-106: “Suggested that it will be essential to develop other combination strategies to produce vaccines against pathogens that present several alternative iron-uptake proteins . ” This is not a sentence.
Lines 116-118: “Whereas type B contains a polysaccharide capsule consisting of galactose, arabinose, and mannose sugar residue), and type F chondroitin or type D heparin/heparin surface. ” This is not a sentence.
Lines 162-164: “The presence of the virulence gene ptfA shows that a positive association with the disease outcome of cattle found that a positive association with the disease outcome in cattle, …” This is redundant and unclear.
Lines 189-191: “The discovery of closely related strains across multiple species suggests transfer of bacteria between host species is considered in the population biology of P.multocida.” This is not a sentence and the meaning is unclear.
Lines 350-351: “... whole cell bacterins, alum precipitated vaccine (APV), and oil - adjuvant vaccine (OAV), and oil adjuvanted vaccine (OAV).” This is redundant.
Lines 429-434: “In bovine pasteurellosis part of the bacteria surface lipoprotein (PmSLP-3) formulated with Montanide ISA 61 fully protective for serogroup B challenge create sustained serum IgG titers that stay for 3 years after administration of two doses and simultaneously can protect serogroup E challenge, similarly vaccines composed of PmSLP (1 and 3) antigens can be a applied and effective solution for preventing HS and BRD related P. multocida infections.” This is largely incomprehensible gibberish.
Line 505: “Antibiotics are ineffective.” This is at least misleading, or untrue to a considerable degree. Antibiotic therapy is often effective if the diagnosis is considered and determined quickly, but clearly has limitations that make vaccination a useful complementary approach.
Response 2: you see the most important mistakes to be corrected, We made revisons one by one you can see the changes made in the final version red labeled once.
line 100: adhesins, toxins, proteins involved in iron-controlled mechanisms and iron acquisition,
line 104 -106 These haemin- and hemoglobin-binding proteins requires other combination strategies to produce vaccines against pathogens that present several alternative iron-uptake proteins [29].
line 116 -118 : In general, P. multocida type A express capsules consisting of hyaluronic acid that are involved in adherence to immune cells [31], whereas type B contains a polysaccharide capsule consisting of galactose, arabinose, and mannose sugar residue, as well as a type F chondroitin or type D heparin/heparin surface [8, 32].
Lines 162-164: The presence of the virulence gene ptfA shows that a positive association with the disease outcome of cattle, found that an important epidemiological marker gene for characterizing P.multocida isolates [40].
Lines 189-191: "The discovery of closely related strains across multiple species suggests bacteria transfer between host species as part of the dynamics, structure, and interaction within P. multocida bacteria [38]".
Lines 350-351: "whole-cell bacterins, alum-precipitated vaccine (APV), and oil-adjuvant vaccine (OAV). Bacterins require frequent vaccination for adequate immunity and can cause shock reactions when dense, while the APV has fewer shock reactions and the OAV has virtually none."
Lines 429-434: " In bovine pasteurellosis, part of the bacteria surface lipoprotein (PmSLP-3) formulated with Montanide ISA 61 fully protects against a serogroup B challenge by creating sustained serum IgG titers that stay for 3 years after administration of two doses and can simultaneously protect against a serogroup E challenge [86]. Similarly, vaccines comprising PmSLP (1 and 3) antigens can be applied as an effective solution for preventing HS- and BRD-related P. multocida infections, indicating that PMSLP is an important vaccine component [87]"
Line 505: "Currently some commonly used antibiotics were developing resistance."
Reviewer 3 Report
Comments and Suggestions for Authors
Dear authors,
The article addresses a relevant review topic for the field of animal health, but I highlight some important points that need to be improved or fixed.
Suggestion - the manuscript has important points that need to be rewritten:
- Title: Write the correct scientific name for the bacterial species;
- Abstract: Keywords – italicize the scientific name;
- Introduction: Italicize the bacterial genus and species, taking care not to confuse disease with bacterial genus (line: 39); use punctuation to separate different bacterial species (line: 44); highlight some writing styles that should be considered in the remainder of the introduction.
- Item 2.1 (Line 120) – line 130 (Figure 1) Suggestion: remove the figure or place it in supplementary material – it does not contribute as is;
- Figure 2 (Line 204) Improve the figure, highlighting virulence genes;
- Figure 3 (Line 206) – standardize the writing of the figure legend; use another resource to display the data, or remove it from the article.
- Item 3.1.1 (Line 213 – 221) – standardize the formatting;
- Item 3.1.2 (Line 249) – place the tables in this item; they were taken out of context, as they were added at the end of another topic; end the sentence with punctuation (line 295).
- Item 3.2.1 (Line 297) – the beginning of the paragraph on line 340 needs to be rewritten;
- Item 3.2.2 (Line 348) – the beginning of the paragraph needs to be rewritten (Line 349);
- Line 363: italicize the bacterial genus;
- Lines 376 to 378: rewrite misspelled words – capitalization and misspellings;
- Item 3.2.5 (Line 463) – italicize the bacterial genus and species (Line 480);
- Line 498 – Cite Table 4 in the corresponding text;
- Item 4: Discuss the topic in more detail;
- General: The manuscript brings together old articles for review, few new articles were addressed; I suggest taking a more up-to-date approach to the subject.
Sincerely,

Author Response
Comment 1: The article addresses a relevant review topic for the field of animal health, but I highlight some important points that need to be improved or fixed.
Suggestion - the manuscript has important points that need to be rewritten:
- Title: Write the correct scientific name for the bacterial species.
- Abstract: Keywords – italicize the scientific name;
- Introduction: Italicize the bacterial genus and species, taking care not to confuse disease with bacterial genus (line: 39); use punctuation to separate different bacterial species (line: 44); highlight some writing styles that should be considered in the remainder of the introduction
Response 1: You brought us very valuable comments and suggestions, and we tried to fix the editorial issues and respond to your comments accordingly. We made overall English revisions by the MDPI English editing author service. In general, you can refer to the overall improvements of the manuscript attached to the final clean manuscript. We have already attached the clean version here on your page. Let's address here one by one.
Title: "Advances in Pasteurella multocida Vaccine Development: From Conventional to Next-Generation Strategies"
Abstract " Antigen composition; Pasteurella multocida; vaccines; virulence gene
Lines 33 and 44 have already been corrected.
Comments 2: Item 2.1 (Line 120) – line 130 (Figure 1) Suggestion: remove the figure or place it in supplementary material – it does not contribute as is;
- Figure 2 (Line 204): Improve the figure, highlighting virulence genes.
- Figure 3 (Line 206) – standardize the writing of the figure legend; use another resource to display the data, or remove it from the article.
- Item 3.1.1 (Line 213 – 221) – standardize the formatting;
Response 2: good suggestion, but we try to see the figures described as an overall pooled proportion collected from so many scholars that shows the association of capsular serotypes and host species (Fig. 1), the association of host species and major virulence gene, and association of capsular serotypes and virulence gene(Fig. 2). This becomes a basis for collecting immunogenic subunits as a guide for vaccine candidacy so that people can refer to that. But the modification and format were corrected; you can see the final version attached below.
Comments 3: Item 3.1.2 (Line 249) – place the tables in this item; they were taken out of context, as they were added at the end of another topic; end the sentence with punctuation (line 295).
- Item 3.2.1 (Line 297) – the beginning of the paragraph on line 340 needs to be rewritten;
- Item 3.2.2 (Line 348) – the beginning of the paragraph needs to be rewritten (Line 349);
- Line 363: italicize the bacterial genus;
- Lines 376 to 378: rewrite misspelled words – capitalization and misspellings;
- Item 3.2.5 (Line 463) – italicize the bacterial genus and species (Line 480);
- Line 498 – Cite Table 4 in the corresponding text;
- Item 4: Discuss the topic in more detail.
Response 3: Thank you for your suggestion, but tables 2 and 3 were narrated overall based on the type of vaccine and respective host species. Section 3.1.2 talks more about active immunization in general; hence, specific types of active immunization and their details were covered by Section 3.2 and the line. So it's better to be at the end of this section.
The other editorial comments were accepted as per your recommendation, and the Section 4, we made improvements.
Overall, you can see on the attached version with track changes still additional modifications were made accordingly
Round 2
Reviewer 2 Report
Comments and Suggestions for Authors
The authors had an editing service revise the manuscript and partially address some of the specific flaws identified as a non-exhaustive list of examples of poor writing, confusion, and lack of clarity in the original manuscript. These changes have improved the manuscript, but not completely resolved deficiencies. I can again provide a handful of specific examples of problems, but I don’t think fixing these will substantially improve the readability or increase the value of the manuscript.
Line 71, “occurring as more or less acute septicemia” - I don’t know what this means.
Line 83, “a reliable pooled protection rate estimate” - I don’t know what this means.
Line 101, “None of them protected in mouse trials under P. multocida experimental infection when inoculated as a single formulation.” In a paragraph about virulence factors, this sentence apparently referencing vaccine protection experiments is poorly introduced, out-of-place, and confusing.
Line 125, “Even though there is a significant difference in proportion, most of the combinations of genes encoding outer membrane proteins, colonization factors, iron acquisition factor, and superoxide dismutase have no basic difference regarding host specificity.” - I don’t know what this means.
Line 181, “these proteins possess diversifying selection within the host” - I don’t know what this means.
I’ll stop there.
Comments on the Quality of English LanguageThe authors had an editing service revise the manuscript and partially address some of the specific flaws identified as a non-exhaustive list of examples of poor writing, confusion, and lack of clarity in the original manuscript. These changes have improved the manuscript, but not completely resolved deficiencies. I can again provide a handful of specific examples of problems, but I don’t think fixing these will substantially improve the readability or increase the value of the manuscript.
Line 71, “occurring as more or less acute septicemia” - I don’t know what this means.
Line 83, “a reliable pooled protection rate estimate” - I don’t know what this means.
Line 101, “None of them protected in mouse trials under P. multocida experimental infection when inoculated as a single formulation.” In a paragraph about virulence factors, this sentence apparently referencing vaccine protection experiments is poorly introduced, out-of-place, and confusing.
Line 125, “Even though there is a significant difference in proportion, most of the combinations of genes encoding outer membrane proteins, colonization factors, iron acquisition factor, and superoxide dismutase have no basic difference regarding host specificity.” - I don’t know what this means.
Line 181, “these proteins possess diversifying selection within the host” - I don’t know what this means.
I’ll stop there.
Author Response
Comment 1: Line 71, “occurring as more or less acute septicemia” - I don’t know what this means.
Response 1: That means it mostly occurred as acute septicemia, but i addition to this, the disease can be manifested in other symptoms that commonly resemble respiratory symptoms. affected organ means (affected lung )
Comment 2: Line 83, “a reliable pooled protection rate estimate” - I don’t know what this means.
Response 2: That means a lot of vaccine trials were made, multiple vaccine types discovered, but still, the overall protection level of all Pasteurella multocida vaccines can not be accurately determined. An additional reason is that vaccines produced everywhere do not cross-protect mostly. So you can not drive a common protection level (protection percentage). Further elaboration was made within the paragraph.
Comment 3: Line 101, “None of them protected in mouse trials under P. multocida experimental infection when inoculated as a single formulation.” In a paragraph about virulence factors, this sentence, apparently referencing vaccine protection experiments, is poorly introduced, out of place, and confusing.
Response 3: "None of them protected." This indicates only for hemin- and hemoglobin-binding proteins, not overall virulence factors. If you go through the whole paragraph, it mentions these proteins only. They have a high contribution to the pathogenesis of the disease, but a vaccine trial sourced from these proteins can not give full protection against challenge infection unless mixed with other immunogens. so that they can not be used solely as vaccine candidates unless mixed with other immunogens.
Comment 4: Line 125, “Even though there is a significant difference in proportion, most of the combinations of genes encoding outer membrane proteins, colonization factors, iron acquisition factor, and superoxide dismutase have no basic difference regarding host specificity.” - I don’t know what this means.
Response 4: This means most genes encoding different virulence factors are found in all hosts; there is no significant difference in the presence of these virulence genes across animal host species. But the proportion of isolation of each virulence gene may vary across different host species. i.e, no single virulence gene is specific to a specific host. For example, the OmpH gene can usually be found 100% in bovines, but may not have a 100% isolation rate in Swine.
Comment 5: Line 181, “these proteins possess diversifying selection within the host” - I don’t know what this means.
Response 5: The virulence genes(example ompH, ompA) that have the same genetic background, i.e, isolated from the same bacterial strain, may have molecular mass heterogeneity across different animal hosts. This suggests a difference in the virulence gene and Host interaction, primarily related to the individual host's immune system, limited to host-specific virulence mechanisms. That is, the majority of strains examined from the different host species could be distinguished from each other either by differences in both their OmpA and OmpH protein sequences and minor proteins, or by differences among their minor proteins. OmpA and OmpH variability generate antigenic diversity, allowing bacteria to evade host antibody recognition. The molecular mass variability of OmpA/OmpH directly affects bacterial virulence. For instance, OmpA allele I strains are more invasive than allele II in bovine kidney cells. At the same time, OmpH-deficient mutants show reduced adhesion to host cells and attenuated virulence in yaks. Generally, this variability allows the bacterium to evade immune detection, colonize diverse host niches, and maintain virulence.
Thank you so much for your detailed technical questions that require a clear description. We tried to answer your questions in this discussion forum, and some modifications were also made to the body of the manuscript attached below with track changes